# Measurement Invariance of the Flourishing Scale among a Large Sample of Canadian Adolescents

**DOI:** 10.3390/ijerph17217800

**Published:** 2020-10-25

**Authors:** Isabella Romano, Mark A. Ferro, Karen A. Patte, Ed Diener, Scott T. Leatherdale

**Affiliations:** 1School of Public Health and Health Systems, University of Waterloo, Waterloo, ON N2L 3G1, Canada; mark.ferro@uwaterloo.ca (M.A.F.); sleatherdale@uwaterloo.ca (S.T.L.); 2Department of Health Sciences, Brock University, St. Catharines, ON L2S 3A1, Canada; kpatte@brocku.ca; 3Department of Psychology, University of Illinois at Urbana-Champaign, Champaign, IL 61820, USA; ediener@illinois.edu

**Keywords:** wellbeing, adolescent wellbeing, adolescent health, measurement, validity

## Abstract

Our aim was to examine measurement invariance of the Flourishing Scale (FS)—a concise measure of psychological wellbeing—across two study samples and by population characteristics among Canadian adolescents. Data were retrieved from 74,501 Canadian secondary school students in Year 7 (2018–2019) of the COMPASS Study and from the original validation of the FS (*n* = 689). We assessed measurement invariance using a confirmatory factor analysis in which increasingly stringent equality constraints were specified for model parameters between the following groups: study sample (i.e., adolescents vs. adults), gender, grade, and ethno-racial identity. In all models, full measurement invariance of the FS across all sub-groups was demonstrated. Our findings support the validity of the FS for measuring psychological wellbeing among Canadian adolescents in secondary school. Observed differences in FS score among subgroups therefore represent true differences in wellbeing rather than artifacts of differential interpretation.

## 1. Introduction

In recent years, there has been increased focus on positive psychology within adolescent health literature. Rather than an emphasis on dysfunction and psychopathology, shifts toward positive psychology are commensurate with shifting dual continua models of mental health and illness [1,2] and newer theories of positive youth development [3,4,5]. Factors grounded in positive mental health and resilience—such as social and psychological wellbeing—have been identified as protective against the development of mental disorders (e.g., anxiety, depression) [6,7]. Flourishing, for example, is a state of overall wellbeing used to describe the presence of mental health [1]. Among adolescents, those who are flourishing are more likely to thrive academically [8] and may be less likely to engage in potentially harmful behaviours including bullying involvement [9], binge drinking [10], and cannabis use [11,12]. Importantly, recognizing adolescence as a developmental period often involving risk-taking and experimentation [13,14], individuals’ overall sense of wellbeing may even buffer the mental health sequelae of such behaviours [11,15].

The secondary school environment presents an opportunity for interventions which aim to foster adolescents’ psychological wellbeing and resilience. Large data systems, such as the COMPASS Study [16] in Canada, serve as valuable infrastructure by which such school-based interventions can be evaluated. However, this work requires robust methods of measuring (and meaningfully quantifying changes in) wellbeing among adolescents. More broadly in Canada, there is an expressed need for population-level indicators of positive mental health and wellbeing in surveillance research, and colleagues have placed effort on finding an appropriate measure [17]. While the measurement of concepts such as wellbeing is inherently challenging [18], several tools exist. The Scale of Positive and Negative Experience (SPANE) and Flourishing Scale (FS), for example, were developed by Diener and colleagues [19] in light of contemporary theories of subjective wellbeing. Whereas researchers have been traditionally concerned with either hedonic (i.e., positive feelings) or eudemonic (i.e., positive functioning) beliefs [20], more recent theorists have argued in favour of an integrative approach to conceptualizing wellbeing [21,22,23] rather than one in which hedonia and eudemonia are considered mutually exclusive. Definitions of flourishing have referred to a state in which hedonic and eudemonic wellbeing are simultaneously present [21,24].

Our current study is concerned specifically with the FS, which consists of eight concise items related to concepts such as autonomy, self-esteem, optimism, and relationships. The FS provides a single score of respondents’ self-perceived psychological wellbeing [19]. Over the last decade, the tool has been validated across several populations (e.g., post-secondary students [25,26,27,28], older adults [29] and with respect to different cultural contexts and languages (e.g., Arabic [30], Chinese [31,32,33], Dutch [34,35], French [36], Greek [37], Persian [38], Portuguese [39,40], Spanish [41,42,43], Urdu [44])). Yet, while the FS has been shown to have strong psychometric properties across adult age groups [26,45], research among adolescents in Canada or elsewhere has been limited. 

One particular question in psychometric research relates to whether a given scale measures the actual construct it was intended to measure (e.g., psychological wellbeing), and whether measurement of the construct is interpreted consistently despite differences in respondents’ characteristics. This property—referred to as measurement invariance—is a prerequisite for meaningfully comparing a measure between groups [46], thus limiting risk of bias as a function of differential interpretation [47]. To our knowledge, no previous studies have sought to test measurement invariance of the FS among adolescents in Canada. Relying on a large, school-based sample of Canadian students in the COMPASS Study [16], the primary objectives of our present study were twofold: first, to test measurement invariance of the FS between our study sample (i.e., adolescents) and Diener’s original study sample (i.e., adults); second, to test measurement invariance of the FS among adolescents by sociodemographic subgroups. These included gender, grade (age), and ethno-racial identity given their relevance as determinants of health and wellbeing among adolescents. Our secondary objective sought to estimate differences in FS scores across adolescent subgroups (i.e., by gender, grade, and ethno-racial identity). 

## 2. Materials and Methods 

### 2.1. Design and Samples

The present study used student-level data from Year 7 (Y_7_ (2018, 2019)) of the COMPASS Study—a large, prospective cohort study (2012–2021) of secondary school students in Canada [16]. Students self-reported various behavioural (e.g., physical activity, substance use) and mental (e.g., depression, anxiety) health indicators using the COMPASS student questionnaire (Cq), administered annually within participating schools during class time. In Y_7_, a total of 74,501 students across 136 schools (8 in Alberta, 15 in British Columbia, 51 in Ontario, 52 in Québec) completed the Cq. Data were collected using active-information, passive-consent data collection protocols [48] that have been approved by the University of Waterloo Office of Research Ethics and participating school boards. Further COMPASS Study details are available elsewhere in print [16] and online (www.compass.uwaterloo.ca). 

Additional data were obtained from the original, multi-site validation study of Diener’s FS [19]. The study sample included 689 adult participants recruited across six post-secondary institutions in the United States and Singapore. Using these data, the FS was shown to have good psychometric properties and strong convergent validity with existing measures of wellbeing [19]. Findings of the original FS validation are available in print [19].

### 2.2. Instrument

The FS [19] was included as a component of the Cq’s mental health module (MH-M; [49,50]). The FS consists of the following statements: (1) I lead a purposeful and meaningful life, (2) my social relationships are supportive and rewarding, (3) I am engaged and interested in my daily activities, (4) I actively contribute to the happiness and wellbeing of others, (5) I am competent and capable in the activities that are important to me, (6) I am a good person and live a good life, (7) I am optimistic about my future, and (8) people respect me. In Diener’s original FS [19], individuals are asked to rate their level of agreement to each statement using a 7-point Likert scale (1 = strongly disagree, 2 = disagree, 3 = slightly disagree, 4 = neither agree nor disagree, 5 = slightly agree, 6 = agree, 7 = strongly agree), where possible sum scores range from 8 to 56 and higher scores indicate greater psychological wellbeing. For the purpose of suitability in large-scale school data collections through COMPASS [49], the FS response options were modified on the Cq to a 5-point Likert scale (1 = strongly agree, 2 = agree, 3 = neither agree nor disagree, 4 = disagree, 5 = strongly disagree) yielding a score between 8 and 40, but where higher scores were indicative of poorer psychological wellbeing. 

We reverse-coded the FS scoring in the Cq MH-M for consistency with Diener’s original FS, and collapsed the strongly agree/agree and strongly disagree/agree Likert response options of the original FS for consistency with the Cq MH-M. As a result of these changes, composite FS scores collected from both samples ranged between 8 and 40 and higher scores indicated greater psychological wellbeing. Internal consistency of the FS was high within the adolescent COMPASS Study sample (α = 0.87) as well as in Diener’s original sample (α = 0.87).

The Cq also allows students to self-report on various sociodemographic characteristics including gender (“male, female” in the Cq and hereafter referred to as “boy, girl”), grade (9,10,11,12, and “other,” which included students in Québec enrolled in Secondaire I and II—equivalent to grades 7 and 8 in other provinces), as well as those enrolled in a secondary school class with no official grade equivalent (e.g., “new immigrant” classes in Québec). Note that there is no grade 12 in Québec. Age and weekly spending/saving money (CAD 0, 1–20, 20–100, 100+, don’t know) were used as a proxy measure of student-level socioeconomic status and part-time employment. Students also reported their ethno-racial identity by selecting from one or more listed identities in the Cq. Using these responses, we re-categorized students as racialized (Black, Asian, Latin American, Indigenous, other, mixed) or non-racialized (white).

### 2.3. Statistical Analysis Strategies

Measurement invariance of the FS was investigated using a four-step procedure in which increasingly stringent equality constraints were specified for model parameters between groups (e.g., adolescents vs. adults; boys vs. girls) within a multiple-group confirmatory factor analysis. In the first step, the configural model imposed no equality constraints on parameters and was the origin for subsequent tests [51]. Configural invariance suggests that the same underlying factor structure is observed between comparison groups. Second, the metric model examined the extent to which the factor loadings for each item were equivalent between groups. Invariant factor loadings are a prerequisite for making valid group comparisons [52]. Third, the scalar model tested for evidence that item intercepts were equivalent [46], so as to verify whether mean differences at the item level are fully explained by mean differences at the factor level. In the final step, the strict model was specified to determine whether the variances of the regression equations for each item were equivalent across groups (i.e., item residuals). Strict invariance is required for defensible item-level comparisons [53]. This systematic approach to adding constraints allows the identification of specific parameters that contribute to model misfit and, ultimately, differences in the interpretation of the latent construct [47]. When model fit is adequate and change in fit indices negligible, equal factor loadings (i.e., metric invariance) suggest that groups attribute the same meaning to the construct; equal factor loadings and intercepts (i.e., scalar invariance) suggest that meaning of the items that comprise the construct is the same between groups; and equal factor loadings, intercepts, and residuals suggest that the explained variance is the same and the construct is measured identically.

We relied on two criteria to establish measurement invariance. The first required adequate model fit at each level of testing. We determined a priori that at least two fit indices (comparative fit index (CFI), square root mean residual (SRMR), or root mean standard error of approximation (RMSEA)) needed to meet established cut-points to declare adequate model fit [54,55,56]. The cut-points were CFI ≥ 0.950; SRMR ≤ 0.080 and RMSEA ≤ 0.080 [46]. The second specified that changes in fit indices (i.e., from the model with fewer equality constraints on parameters to the more constrained model) must not exceed established cut-points. We determined a priori that of ΔCFI, ΔSRMR, or ΔRMSEA scores, at least two needed to meet this criterion to establish measurement invariance at any given level of testing. The cut points for change in model fit indices were ΔCFI ≤ −0.010, ΔSRMR ≥ 0.030, or ΔRMSEA ≥ 0.015 [57]. Given that χ^2^ goodness-of-fit and Δχ^2^ are highly influenced by sample size, we did not rely on them as indices of model fit.

If model fit was inadequate at the configural level, we reviewed modification indices to identify potential correlations between like items that could be specified to improve model fit. Where measurement invariance at a given level of testing was not established (i.e., substantial worsening of model fit), we reviewed modification indices and identified constraints on relevant non-invariant parameters that could be removed to improve model fit. We then tested the prespecified model against the less constrained model and computed change scores again. This approach, known as partial invariance, argues that only a subset of model equivalent parameters is needed for substantive analyses between groups [58].

Measurement invariance testing was conducted with robust standard errors [59,60] using Mplus version 6.11 (Muthén & Muthén, Los Angeles, CA, USA) [61]. Additional descriptive (t and one-way ANOVA tests) and parametric (mixed linear regression) analyses were conducted in SAS version 9.4 (SAS Institute Inc., Cary, NC, USA) [62]. Adjusted β-estimates (controlling for province and weekly spending/saving money) were reported alongside 95% confidence intervals. We calculated the intra-class correlation coefficient (ICC) explaining potential variation in students’ FS score as a result of school-level clustering (ICC_FS_ = 0.033). Although we detected a marginal amount of within-school variation, we proceeded to adjust for students’ clustering by schools within all measurement invariance tests and regression modelling. We used the full information maximum likelihood function in Mplus and SAS (PROC MIXED) to preserve cases with missing data [63].

## 3. Results

### 3.1. Sample Characteristics

Among COMPASS Y_7_ participants (*N* = 74,501), the mean FS score was 32.19 (SD = 5.72). Students’ sum scores ranged from 8 to 40, and the median FS score was 32; 41% (*N* = 28,553) of students reported FS scores below the mean. Half of students (50%) were girls, and 31% reported a racialized identity. Students’ mean age was 15 years (SD: 1.5 years, range: 12 to 19 years) and roughly 81% were within grades 9–12. The majority of students were from Ontario (41%) or Québec (30%). Student characteristics are presented with mean FS score in Table 1; FS total and item-level means are further shown by subgroup in Appendix A. Refer to Table 2 for FS score norms (as percentile rankings) among students in our sample.

A total of 6% (*n* = 4,486) of COMPASS Y_7_ students were missing FS scores. Some item non-response was also observed across measures of gender (1%, *n* = 829), grade (1%, *n* = 818), ethno-racial identity (<1%, *n* = 662), and spending money (<2%, *n* = 1079). Additional analyses predicting missingness in the FS by student characteristics are available in Appendix B. Students were more likely to have missing FS scores if they were boys or racialized; log-odds of missingness decreased with increasing school grade.

### 3.2. Measurement Invariance by Sample

Given that a total composite score is used for the FS, we proceeded directly to fitting the single-factor model in the confirmatory factor analysis. Fit of the one-factor model was good among the adolescent (χ^2^_20_ = 7,067.17; CFI = 0.959; SRMR = 0.028; RMSEA = 0.070 [90% CI, 0.068–0.071]) and adult samples (χ^2^_20_ = 56.97; CFI = 0.966; SRMR = 0.031; RMSEA = 0.052 [90% CI, 0.036–0.068]), independently and in the configural model (Table 3). As shown in Table 3, equality constraints placed on the factor loadings (metric model) did not substantially worsen model fit: ΔCFI = 0.000 and ΔRMSEA = −0.006. Similar results were found when constraining item intercepts (scalar model: ΔCFI = −0.006; ΔRMSEA = 0.000) and residuals (strict model: ΔCFI = −0.001; ΔRMSEA = −0.003).

### 3.3. Measurement Invariance by Gender, Grade, and Ethno-Racial Identity

Having established full measurement invariance between adolescents and adults, invariance across gender, grade, and ethno-racial identity was tested among the adolescent sample (Table 4). The configural models (no equality constraints) demonstrated excellent fit to the data for gender (χ^2^_40_ = 4777.31; CFI = 0.994; SRMR = 0.026; RMSEA = 0.059 [90% CI, 0.057–0.061]), grade (χ^2^_40_ = 1622.78; CFI = 0.979; SRMR = 0.024; RMSEA = 0.058 [90% CI, 0.056–0.061]), and ethno-racial identity (χ^2^_40_ = 6270.29; CFI = 0.990; SRMR = 0.029; RMSEA = 0.065 [90% CI, 0.064–0.067]). Constraints imposed at the metric, scalar, and strict levels did not result in substantial worsening of model fit; full gender, grade, and ethno-racial identity invariance was demonstrated.

### 3.4. Flourishing Scale Scores Among COMPASS Y_7_ Students

Results from a mixed linear regression model predicting a one-point increase in the FS, on average, are presented in Table 5 (Model I). Lower average FS scores were present among girls (β = −0.88, *p* < 0.0001) compared to boys, as well as students who reported a racialized identity (β = −0.83, *p* < 0.0001) compared to those who did not. Compared to students in grade 9, average FS scores were lower in grades 10 (β = −0.35, *p* < 0.0001), 11 (β = −0.54, *p* < 0.0001), and 12 (β = −0.63, *p* < 0.0001) but higher for those in the “other” grade category (β = 0.92, *p* < 0.0001). All estimates are adjusted for province and weekly spending money.

## 4. Discussion

Using recent data collected from a large sample of secondary school students enrolled in the COMPASS Study, our findings support the validity of the FS among the Canadian adolescent population. We demonstrated full measurement invariance between our sample and the adult sample among whom the FS was originally validated [19], thereby confirming that the FS in fact measures the same construct (i.e., psychological wellbeing) among adolescents. Additionally, we identified strict invariance in FS measurement across gender, grade, and ethno-racial identity. This finding confirms that statistical differences observed between student subgroups within the COMPASS Study represent true differences, rather than artifactual differences in interpretation of the FS. Given the importance of psychological wellbeing to adolescent health and strong psychometric properties demonstrated by the FS, future youth-focused surveillance research should strongly consider incorporating the FS as a measure of youth wellbeing.

While the FS has been previously validated across a number of populations and languages, our present work continues to fill important gaps within the existing literature. Our study is the first to test measurement invariance with the sample of participants used in the original development and validation of the FS. To our knowledge, the COMPASS Study also represents the largest sample in which FS validation has been investigated, followed by a study from colleagues in New Zealand who relied on a nationally representative sample of 10,009 adults [45]. With two exceptions, the majority of existing FS validation studies have focused solely on adult populations and/or university students; Singh, Junnarker, and Jaswal found good fit of a single-factor structure for the FS items among adolescents in India [64], as did Duan and Xie among 12–17-year-old adolescents in China [32]. Findings from these studies may not generalize to Canadian adolescents. 

Upon establishing measurement invariance across gender, grade, and ethno-racial identity among students in the COMPASS Study, further extending the validity of the FS, we examined differences in FS scores across these subgroups. First, we identified lower levels of psychological wellbeing among girls than boys. This finding is consistent with existing research indicating relatively poor wellbeing among adolescent girls compared to boys [65,66,67] yet is inconsistent with findings from a smaller Canadian study in which female undergraduate students scored higher on the FS than males once validity was established [26]. Other findings have shown no differences by sex among youth [32]. While previous studies of the FS have demonstrated measurement invariance by sex and gender [27,32,37], few have gone on to test for the presence of differences that describe gendered experiences of wellbeing. Our current findings may reflect the ways in which girls are disproportionately impacted by socio-cultural norms and pressures during adolescence, experiences of which may influence their wellbeing [68].

We also found a pattern of decreasing psychological wellbeing with increasing secondary school grade. Consistent with other school-based and youth health literature, positive indicators of wellbeing generally appear to decrease with adolescent age toward young adulthood [65,66]–perhaps as a function of factors including perceived academic and social stress and pubertal and psychosocial development. As grade and age are highly correlated among students in the COMPASS Study, we chose to adopt grade in our analyses as a proxy for age to improve the interpretability of our findings for school-based knowledge users. Given measurement invariance of the FS by grade was established within our sample ranging in age from 12 to 19 years, we highlight an ability to detect age-related differences in psychological wellbeing among adolescents using the FS. 

After confirming measurement invariance by ethno-racial identity among COMPASS Y_7_ students, we found that racialized students reported lower average psychological wellbeing compared to non-racialized students. An existing body of knowledge recognizes racial and ethnic discrimination as linked to various health outcomes and disparities [69,70] as well as self-perceived and psychological wellbeing [71,72,73]. Here, we provide evidence of measurement invariance of the FS by ethno-racial identity. It is important to note that due to sample size restrictions, we were unable to assess wellbeing across racial or ethnic identity groups as self-reported by students. However, our findings continue to highlight the ways in which individuals’ experiences of wellbeing may be differentially impacted by socio-political and systemic processes of racism, discrimination, and stigmatization—even among adolescents in Canada. 

### 4.1. Strengths and Limitations

These findings are primarily strengthened by the nature of our data. The COMPASS Study represents the largest school-based sample of adolescents in Canada—data from whom are collected across several provinces (Alberta, British Columbia, Ontario, Québec) through a hierarchical design. Despite our inability to test convergent validity of the FS with other measures of wellbeing, we were uniquely able to use existing data originally collected by Diener and colleagues [19] to validate the FS; thus allowing us to establish measurement invariance of the FS across adolescents and adults. Analytically, our findings are further strengthened by (1) our use of full-information maximum likelihood to handle missing data, rather than relying on complete-case analysis and (2) our multi-level modelling approach to control for variability due to school-level clustering.

We note some limitations. First, these self-reported data are not nationally representative, and the generalizability of our findings to all adolescents in Canada is thus limited. However, the COMPASS Study relies on purposive sampling procedures that contribute to our large sample size, and use of active-information, passive-consent data collection protocols helps mitigate bias introduced by students’ self-reporting [48,74,75,76,77,78]. Second, while we assessed differences in wellbeing by gender and ethno-racial identity, we were not able to account for non-binary gender identities and heterogeneity among racialized groups. Moreover, we did not investigate interactions across sub-groups of gender, grade, and ethnicity/race; intersectional analyses are necessary to further understand differential experiences of wellbeing. Third, our findings are only relevant to secondary school-aged adolescents in Canada and not to younger students. Future studies are needed to investigate the applicability of the FS for measuring wellbeing among primary and elementary school-aged children.

### 4.2. Implications

These findings have practical implications for health researchers and practitioners interested in using the FS to measure wellbeing among adolescents. We have shown here that the FS can be used as a valid tool for assessing and monitoring the psychological wellbeing of Canadian adolescents, as it not only measures the construct as originally intended but does so consistently despite differences in students’ gender, grade, and ethno-racial identity. Data collected using the FS can inform targeted interventions meant to promote the overall psychological wellbeing of adolescents within secondary school settings. Notably, our findings are situated within the secondary school context, thus highlighting the potential utility of findings in demonstrating the FS as an indicator of successful school-based interventions and programs. Through robust data systems such as the COMPASS Study, the impact of these interventions and policies (i.e., at the school-, provincial-level, etc.) can be evaluated in real time as natural experiments [16,79].

## 5. Conclusions

In summary, our findings demonstrate full measurement invariance of the FS between study samples, and across the adolescents’ gender, grade, and ethno-racial identity. These findings further support the validity of the FS for measurement of psychological wellbeing among Canadian adolescents in the secondary school context. Further, this study provides evidence that efforts to improve psychological wellbeing should especially consider the needs of adolescent girls, those in older secondary school grades, and racialized students. Using existing data systems, such as the COMPASS Study, Canadian programs and interventions that target students’ wellbeing can be evaluated as robust yet feasible natural experimental studies.

## Figures and Tables

**Table 1 ijerph-17-07800-t001:** Descriptive comparisons of Flourishing Scale scores by student characteristics, among Y_7_ COMPASS Study (2018–2019) participants.

		Flourishing Scale ^1^ Score		
Measure	*n* (%)	Mean (SD)	*F*, *t*	*p*
**Gender**				
Boys *(ref.)*	37,126 (50.4)	32.64 (5.60)	19.80	<0.0001
Girls	36,546 (49.6)	31.78 (5.75)		
**Grade**				
9 *(ref.)*	17,294 (23.5)	32.14 (5.67)	362.47	<0.0001
10	17,201 (23.3)	31.86 (5.63)		
11	15,940 (21.6)	31.77 (5.69)		
12	9357 (12.7)	31.29 (5.86)		
Other ^2^	13,891 (18.9)	33.83 (5.43)		
**Ethno-Racial Identity**				
Non-racialized *(ref.)*	51,017 (69.1)	32.63 (5.53)	30.88	<0.0001
Racialized	22,822 (30.9)	31.18 (6.00)		
**Weekly Spending Money**				
CAD 0 or “don’t know” *(ref.)*	24,669 (33.6)	31.83 (5.93)	51.98	<0.0001
CAD 1–20	17,744 (24.2)	32.26 (5.55)		
CAD 21–100	16,793 (22.9)	32.51 (5.41)		
CAD 100+	14,216 (19.3)	32.35 (5.88)		
**Province**				
Alberta *(ref.)*	3301 (4.4)	31.63 (5.79)	664.79	<0.0001
British Columbia	10,402 (14.0)	30.95 (5.75)		
Ontario	30,675 (41.2)	31.54 (5.82)		
Québec	30,123 (40.4)	33.31 (5.41)		
**Total sample**	74,501 (100.0)	32.19 (5.72)		

Note*. ref.* = reference category; SD = standard deviation. ^1^ Higher scores indicate greater psychological wellbeing. ^2^ Includes Secondaire I and II in Québec.

**Table 2 ijerph-17-07800-t002:** Flourishing Scale score norms among Y_7_ COMPASS Study (2018–2019) participants, by percentile.

Percentile Rank	Flourishing Scale ^1^ Score
0	8
1	15
5	22
10	25
25	29
50	32
75	37
90	40
95	40
99	40
100	40

Note. Flourishing Scale scores range from 8 to 40; percentile rankings for scores not shown can be approximated by interpolation. ^1^ Higher scores indicate greater psychological wellbeing.

**Table 3 ijerph-17-07800-t003:** Measurement invariance of the Flourishing Scale across Y_7_ COMPASS Study (2018–2019) adolescent and original adult samples.

Parameter	χ^2^ (df)	CFI	SRMR	RMSEA (90% CI)	Δχ2 (df)	ΔCFI	ΔSRMR	ΔRMSEA
Configural	7447.84 (40)	0.959	0.028	0.071 (0.070, 0.073)	-	-	-	-
Metric	7463.04 (48)	0.959	0.082	0.065 (0.064, 0.066)	8.2 (8)	0.000	0.054	−0.006
Scalar	8528.16 (55)	0.953	0.101	0.065 (0.064, 0.066)	687.6 (7)	−0.006	0.019	0.000
Strict	8785.83 (63)	0.952	0.094	0.062 (0.060, 0.063)	165.4 (8)	−0.001	−0.007	−0.003

*Note. df* = degrees of freedom; CFI = comparative fit index; SRMR = square root mean residual; RMSEA = root mean standard error of approximation; CI = confidence interval.

**Table 4 ijerph-17-07800-t004:** Measurement invariance of the Flourishing Scale across characteristics among Y_7_ COMPASS Study (2018–2019) participants.

Parameter	χ^2^ (df)	CFI	SRMR	RMSEA (90% CI)	Δχ2 (df)	ΔCFI	ΔSRMR	ΔRMSEA
**Gender** ^1^								
Configural	4777.31 (40)	0.994	0.026	0.059 (0.057, 0.060)	-	-	-	-
Metric	5496.54 (48)	0.993	0.083	0.057 (0.056, 0.059)	323.2 (8)	−0.001	0.057	−0.002
Scalar	6277.60 (55)	0.992	0.082	0.057 (0.056, 0.059)	409.5 (7)	−0.001	−0.001	0.000
Strict	6396.81 (63)	0.991	0.081	0.054 (0.052, 0.055)	57.8 (8)	−0.001	−0.001	−0.003
**Grade** ^2^								
Configural	1622.78 (40)	0.979	0.024	0.058 (0.056, 0.061)	-	-	-	-
Metric	2058.53 (48)	0.973	0.094	0.059 (0.057, 0.061)	263.7 (8)	−0.006	0.070	0.001
Scalar	2364.05 (55)	0.969	0.094	0.059 (0.057, 0.061)	266.8 (7)	−0.004	0.000	0.000
Strict	2683.41 (63)	0.965	0.098	0.057 (0.056, 0.059)	159.4 (8)	−0.004	0.004	−0.002
**Ethno-Racial Identity** ^3^								
Configural	6270.29 (40)	0.990	0.029	0.065 (0.064, 0.067)	-	-	-	-
Metric	7413.03 (48)	0.988	0.108	0.065 (0.064, 0.066)	474.8 (8)	−0.002	0.079	0.000
Scalar	7604.73 (55)	0.987	0.108	0.061 (0.060, 0.063)	104.0 (7)	−0.001	0.000	−0.004
Strict	7958.49 (63)	0.987	0.114	0.059 (0.058, 0.060)	145.6 (8)	0.000	0.006	−0.002

*Note. df =* degrees of freedom; CFI = comparative fit index; SRMR = square root mean residual; RMSEA = root mean standard error of approximation; CI = confidence interval. ^1^ Boys vs. girls. ^2^ Grade 12 vs. 9. ^3^ Racialized vs. non-racialized.

**Table 5 ijerph-17-07800-t005:** Mixed linear regression model predicting Flourishing Scale scores among Y_7_ COMPASS Study (2018–2019) participants.

	Model I	
Measure	β	(95% CL)
**Gender**		
Boys *(ref.)*	0.00	
Girls	−0.88 ***	−0.88, −0.88
**Grade**		
9 *(ref.)*	0.00	
10	−0.35 ***	−0.35, −0.35
11	−0.54 ***	−0.54, −0.54
12	−0.63 ***	−0.63, −0.62
Other ^1^	0.92 ***	0.91, 0.93
**Ethno-Racial identity**		
Non-racialized *(ref.)*	0.00	
Racialized	−0.83 ***	−0.83, −0.82

Model I: Predicts a one-point increase, on average, in Flourishing Scale score. Note. *ref.* = reference category; CL = confidence limit. All estimates are adjusted for province and weekly spending money. ^1^ Includes Secondaire I and II in Québec. *** *p* < 0.001.

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
