# Peer review of "Measurement Invariance of the Flourishing Scale among a Large Sample of Canadian Adolescents"

_ijerph, 2020, doi:10.3390/ijerph17217800_

Round 1
Reviewer 1 Report
Referee’s report on
Measurement invariance of the Flourishing Scale among a large sample of Canadian adolescents
Summary
This article uses data collected from a large sample of secondary school students enrolled in the COMPASS study to find support for the effectiveness of FS in the Canadian adolescent population, demonstrating complete measurement invariance between the sample and the initially validated adult sample, thus confirming that FS actually measures the same structure in adolescents. These findings have practical implications for health researchers and practitioners interested in using the FS to measure wellbeing among adolescents.
General comments
The topic of this paper is interesting and it will make a lot of contribution in related research field, and very ambitious, Thus, from that perspective the paper certainly deserves publication in Environmental Research and Public Health.
But the article is not dealt with in details, with a clear methodology, which makes that in the end it is extremely difficult to understand in the evidence of any given behavior or attitude.
Due to the lack of description of algorithm processing based on sample data and chart description of method, the current paper needs to be carefully revised to achieve this purpose. Due to the lack of clear algorithm description, the paper is in confusion. In the current form, although it explains the theoretical basis of this study, I think it will be helpful to further discuss the algorithm or method of specific data processing in detail. To do this, I comment on those below.
1 Can the description in the introduction of materials and methods add data or chart descriptions, such as (2.2 can chart compares or illustrations be added to the article to make it easier to read.2.3 A detailed description of the algorithm method should be added to make the article clearer). The author only mentions concepts. It is necessary to briefly indicate their relationship with data or icons, as well as their practical opinions in decision-making
2 The chart in the result description should be reformatted to increase readability.
Recommendation
The ideas in the paper merit publication in Environmental Research and Public Health. but the authors should revise the manuscript to address the general and specific issues mentioned above.
References
Michelle Lambert, Theresa Fleming, Shanthi Ameratunga, Elizabeth Robinson, Sue Crengle, Janie Sheridan, Simon Denny, Terryann Clark & Sally Merry (2014) Looking on the bright side: An assessment of factors associated with adolescents’ happiness, Advances in Mental Health, 12:2, 101-109
Teresa Freire, Gabriela Ferreira. (2018) Health-related quality of life of adolescents: Relations with positive and negative psychological dimensions. International Journal of Adolescence and Youth 23:1, pages 11-24.
Shaojian Qu, Yongyi Zhou, Yulin Zhang, MIM Wahab, Guang Zhang, Yuanyuan Ye. Optimal strategy for a green supply chain considering shipping policy and default risk, Computers and Industrial Engineering, 2019, 131:172-186.
MacDonald FJ, Bottrell D, Johnson B. Socially transformative wellbeing practices in flexible learning environments: Invoking an education of hope. Health Education Journal. 2019;78(4):377-387.
Kubiszewski I, Zakariyya N, Jarvis D. 2019. Subjective wellbeing at different spatial scales for individuals satisfied and dissatisfied with life. PeerJ 7:e6502
Ji Y., Qu S.j., Wu Z., Liu Z.m.. A Fuzzy-Robust Weighted Approach for Multicriteria Bilevel Games, IEEE Transactions on Industrial Informatics, 2020, 16(8): 5369-5376.
Reviewer 2 Report
Comments to the Author
This manuscript, titled “Measurement Invariance of the Flourishing Scale among a Large Sample of Canadian Adolescents,” examined measurement invariance of the Flourishing Scale (Diener et al., 2010) in a sample of Canadian adolescents. The results showed that the scale had full measurement invariance between adolescents and adults as well as across gender, grade, and ethno-racial identity in the adolescent sample. The manuscript is well-written, and the statistical analyses were sophisticated and appropriate. Tables 3 and 4 clearly summarize the main findings, which are helpful in understanding the findings. To further improve the manuscript, clarity is needed in several parts of the manuscript.
- Section 1 (Introduction) should provide a clearer definition of flourishing. The brief definition of flourishing (lines 50-51) should be lengthened with more details, and it should appear in the first paragraph of the section.
- The secondary objective needs clarity. In particular, it is not clear what “with adjustment for confounding” means (lines 71-71).
- A brief explanation of why gender, grade, and ethno-racial identity were chosen would be helpful.
- The definitions of racialized and non-racialized identity should be provided (lines 117, 172).
- Section 2.3 heading should be replaced with “Statistical Analysis Strategies.”
- There seems to be an extra comma on line 137.
- ∆CFI, ∆SRMR, and ∆RMSEA involved comparisons between a less constrained and a more constrained model (line 144). It would be helpful to explain what a less constrained model and a more constrained model mean.
